# Bioactives in Oral Nutritional Supplementation: A Pediatric Point of View

**DOI:** 10.3390/nu16132067

**Published:** 2024-06-28

**Authors:** Nicola Cecchi, Roberta Romanelli, Flavia Ricevuti, Maria Grazia Carbone, Michele Dinardo, Elisabetta Cesarano, Alfredo De Michele, Giovanni Messere, Salvatore Morra, Armando Scognamiglio, Maria Immacolata Spagnuolo

**Affiliations:** 1Clinical Nutrition Unit, A.O.R.N. Santobono-Pausilipon Children’s Hospital, 80129 Naples, Italy; n.cecchi@santobonopausilipon.it (N.C.); robe.romanelli@gmail.com (R.R.);; 2Department of Pediatrics, University Federico II of Naples, Via Sergio Pansini 5, 80131 Naples, Italy; mariaimmacolata.spagnuolo@unina.it

**Keywords:** pediatric, oral nutritional supplements, enteral nutrition, immunonutrition, fibers, inflammatory, immunity, clinical nutrition, bioactive compounds

## Abstract

Background: Oral nutritional supplements (ONSs) are crucial for supporting the nutritional needs of pediatric populations, particularly those with medical conditions or dietary deficiencies. Bioactive compounds within ONSs play a pivotal role in enhancing health outcomes by exerting various physiological effects beyond basic nutrition. However, the comprehensive understanding of these bioactives in pediatric ONSs remains elusive. Objective: This systematic narrative review aims to critically evaluate the existing literature concerning bioactive compounds present in oral nutritional supplements from a pediatric standpoint, focusing on their types, sources, bioavailability, physiological effects, and clinical implications. Methods: A systematic search was conducted across the major academic databases, including PubMed, Scopus, and Web of Science, employing predefined search terms related to oral nutritional supplements, bioactives, and pediatrics. Studies published between 2013 and 2024 were considered eligible for inclusion. Data extraction and synthesis were performed according to the PRISMA guidelines. Results: The initial search yielded 558 of articles, of which 72 met the inclusion criteria. The included studies encompassed a diverse range of bioactive compounds present in pediatric ONS formulations, including, but not limited to, vitamins, minerals, amino acids, prebiotics, probiotics, and phytonutrients. These bioactives were sourced from various natural and synthetic origins and were found to exert beneficial effects on growth, development, immune function, gastrointestinal health, cognitive function, and overall well-being in pediatric populations. However, variations in bioavailability, dosing, and clinical efficacy were noted across different compounds and formulations. Conclusions: Bioactive compounds in oral nutritional supplements offer promising avenues for addressing the unique nutritional requirements and health challenges faced by pediatric populations. However, further research is warranted to elucidate the optimal composition, dosage, and clinical applications of these bioactives in pediatric ONS formulations. A deeper understanding of these bioactive compounds and their interplay with pediatric health may pave the way for personalized and effective nutritional interventions in pediatric clinical practice.

## 1. Introduction

Oral nutritional supplements (ONSs) are part of the nutritional products for medical nutrition therapy, designed to address specific health conditions and improve overall health and well-being. These products are used under the guidance of healthcare professionals and can include a range of specialized foods, supplements, and enteral or parenteral nutrition solutions. Enteral nutrition (EN) products designed to deliver both macronutrients and/or micronutrients encompass various forms of nutritional support involving the utilization of “Nutritional Products for Medical Nutrition Therapy” as defined by the 2017 European Society for Clinical Nutrition and Metabolism (ESPEN) guidelines [1] on the definitions and terminology of clinical nutrition.

These supplements find extensive application in health settings, particularly for individuals facing challenges in meeting their nutritional needs solely through their oral dietary intake [2].

ONSs are designed to complement the diet of individuals who can eat orally but need extra nutrition: provide balanced nutrition, high protein content for muscle maintenance, and high calories for weight gain or maintain.

ONSs might be recommended for a brief duration during acute illness or for individuals managing long-term chronic conditions and are frequently prescribed or recommended for those at risk of malnutrition. The purpose of ONSs is to supplement nutritional intake, and accompanying guidance on enhancing oral intake should be offered concurrently. Evidence from studies involving adults indicates that the use of ONSs is linked to a decreased length of stay (LOS) [3], lower inpatient episode costs [4], reduced complication rates [5], alleviation of depressive symptoms, lower readmission rates, and enhanced recovery of lean body mass. In pediatrics, nutritional support is crucial for the optimal growth and development of children, and ONSs play a pivotal role in addressing nutritional deficiencies and promoting overall well-being. Furthermore, ONSs, looking at them from the side of children, do not instill that fear of the classic pharmacological/therapeutic treatment of drugs.

ONSs come in diverse formats, such as convenient ready-to-drink beverages (available in milk or juice varieties), powders ideal for meal integration, and delectable dessert-style puddings. Certain products offer a high-fiber variant designed to support optimal bowel function. Additionally, there are products with a denser consistency specifically crafted for individuals dealing with dysphagia. For patients who can fulfill their protein needs but struggle to meet their energy requirements, fat emulsion shots can offer a concentrated source of energy. Table 1 [6] shows the classifications of oral nutritional supplements, clinical indications, and potential advantages linked to the use of these products.

In recent years, researchers have investigated the potential benefits of incorporating bioactive molecules into ONS formulations to address specific nutritional needs in children, so the aim of this paper is to discuss the various aspects of ONSs in pediatric nutrition, emphasizing oral supplements enriched with specific functional molecules with specific physiological activities that extend beyond their nutritional value. These molecules exhibit various bioactivities, including antioxidant, anti-inflammatory, and immunomodulatory properties. Understanding the mechanisms of action of these molecules is crucial for elucidating their potential benefits in pediatric nutrition. This review aims to explore the potential advantages of incorporating biocompounds into ONSs for pediatric use.

## 2. Materials and Methods

This systematic review followed the PRISMA (Preferred Reporting Items for Systematic Reviews and Meta-Analyses) guidelines to ensure comprehensive and transparent reporting of the methods and findings. A PRISMA flow diagram was used to depict the study selection process (Figure 1).

In most cases, systematic reviews culminate in either a statistical (quantitative) or narrative (qualitative) summary of findings. Given the nature of the research questions and designs involved in this review, a narrative review approach was employed to synthesize the data. Narrative reviews are particularly useful for exploring studies that investigate the effects of interventions, the factors influencing the implementation of interventions, the needs and preferences of specific population groups, and the causes of particular social and health problems.

### 2.1. Data Sources and Study Selection

Comprehensive searches were performed across three major academic databases: PubMed, Scopus, and Web of Science. The search strategy utilized a combination of keywords and Medical Subject Headings (MeSH) terms, including “oral nutritional supplements”, “bioactives”, “pediatrics”, “children”, “nutritional support”, and “health outcomes”. Boolean operators (AND and OR) were employed to refine the search queries. The search was restricted to studies published from January 2013 to April 2024. The inclusion criteria encompassed studies published in English, involving pediatric populations aged 0–18 years, investigating bioactive components within oral nutritional supplements, and reporting outcomes related to efficacy, safety, or both. The exclusion criteria comprised studies not relevant to the pediatric population, non-English publications, and those lacking full-text availability. Two independent reviewers screened in blind titles, abstracts, and full texts of identified studies for eligibility, with disagreements resolved through discussion or consultation with a third reviewer.

### 2.2. Data Extraction

Data were independently extracted by two reviewers (RR and FR) using a standardized form using software for systematic reviews [7]. The extracted information included study characteristics (author, year, country, and study design), population details (age and sample size), types of bioactive compounds, sources of bioactives (natural or synthetic), the outcomes measured, and key findings related to the effects of bioactives on the specified outcomes.

### 2.3. Results

The initial search identified 558 articles, of which 72 met the inclusion criteria. These studies examined a wide range of bioactive compounds present in pediatric ONS formulations, including vitamins, minerals, amino acids, prebiotics, probiotics, and phytonutrients. These bioactives, sourced from both natural and synthetic origins, demonstrated beneficial effects on growth, development, immune function, gastrointestinal health, cognitive function, and overall well-being in pediatric populations. However, there were noted variations in bioavailability, dosing, and clinical efficacy across different compounds and formulations.

## 3. Dietary Fibers as Bioactive Compounds in Oral Nutritional Supplements

Dietary fibers encompass a diverse array of substances crucial for gut health and a healthy microbiome, but not all oral nutritional supplements contain fiber. Why? Many children with digestive diseases or conditions (e.g., strictures or recurrent small bowel obstructions) need to avoid fiber in the short- or long-term. Oral nutritional supplements with added fiber can be a valuable tool in promoting the overall health of children, especially those facing nutritional challenges. The American Gastroenterology Association (AGA) and the European Society of Pediatric Gastroenterology, Hepatology, and Nutrition (ESPGHAN) recently published different evidence-based recommendations.

This paragraph reviews the physiological effects of dietary fiber, the impact of fiber on gastrointestinal health, and the potential synergies between fiber and essential nutrients in ONSs in the pediatric population. Additionally, we discuss the challenges and opportunities associated with formulating ONSs with added fiber, along with emerging research directions in this evolving field. By delving into the scientific and clinical aspects of ONSs with added fiber, this paper aims to contribute to the growing body of knowledge surrounding nutritional and dietary supplementation.

The importance of the dietary fiber children consume is just as significant as the quantity they intake. The majority of oral nutritional supplements, including those with added fiber, are still considered to be low-fiber and below the recommended amount per day. Fiber is crucial for preserving the structure and function of the gut microbiome, promoting overall host health [8]. The latest agreement suggests that dietary fiber consists of carbohydrate polymers that undergo neither digestion nor absorption in the human intestine. Instead, they proceed to the large bowel, where the colonic microbiota partially or completely ferment them.

Dietary fibers and whole grains consist of a unique blend of bioactive components, incorporating plant non-starch polysaccharides like cellulose, pectin, gums, hemicelluloses, and β-glucans, as well as vitamins, minerals, phytochemicals, and antioxidants. Natural fibers refer specifically to dietary fibers naturally occurring in the cell wall or inner layer of diverse edible plants, fruits, vegetables, cereals, nuts, pulses, and seaweed. Furthermore, fibers can be derived from plants or food and undergo modifications through different processing techniques [9]. The physicochemical properties of various dietary fibers, including solubility, viscosity, and fermentability, exhibit significant variations based on their origin and processing. These factors play a crucial role in determining the functional characteristics and clinical efficacy of dietary fibers. Even with advancements in comprehending these connections, numerous clinical inquiries persist, such as determining the optimal dosage, type, and source of fiber essential for effectively managing clinical symptoms and preventing gastrointestinal disorders [10]. Despite endeavors to promote healthy eating habits, certain factors, such as picky eating, medical conditions, or lifestyle constraints, may impede some children from meeting their nutritional needs. In such instances, oral nutritional supplements (ONSs) serve as a valuable tool to bridge nutritional gaps. The incorporation of fiber into ONSs for children is particularly noteworthy due to its contributions to digestive health, satiety, and disease prevention.

In studies evaluating the effects of fibers in children, no serious adverse events have been reported. Since fibers are resistant to digestion and are not absorbed in the human small intestine, partial or complete fermentation occurs in the large intestine, leading to flatulence, diarrhea, abdominal distention, and discomfort. These effects vary based on the quantity and type of fibers, other food components, the gut microbiota, and individual response and sensitivity [11,12,13,14].

### 3.1. Prebiotic Fibers in Oral Nutritional Supplementations for Children

Non-digestible fermentable carbohydrates, or fermentable or “prebiotic” fibers, are a heterogeneous group of polysaccharides and oligosaccharides constituting dietary fiber. These compounds, therefore, could be defined as “nourishment for the intestinal microbiota”, which can be modulated and regulated, thus promoting microbial diversity, which in turn affects the regulation of energy metabolism, intestinal homeostasis, and the production of metabolites that actively participate in immune responses. Currently, the literature extensively documents only resistant oligosaccharides, such as fructans and galactans, as accepted prebiotics. Other fibers are regarded as potential prebiotics or possess prebiotic potential, while some seem to lack a discernible prebiotic effect on humans [15]. Among the fermentable fibers, galacto-oligosaccharides (or GOSs, present in breast milk and yogurts) and fructo-oligosaccharides (or FOSs, present in *Liliaceae*, bananas, rye, and wheat) are especially well known, with a prebiotic role mainly on the colon microbiota [16]. These functional food components, as shown, may be naturally present in plant-based foods or produced synthetically through the enzymatic conversion of sugars [17]. Functional oligosaccharides, including galacto-oligosaccharides (GOSs), fructo-oligosaccharides (FOSs), xylo-oligosaccharides, isomaltose, and human milk oligosaccharides, reach the colon and act with unique physiological functions: preventing intestinal obstruction or diarrhea, regulating intestinal flora, or promoting the proliferation of bifidobacteria, positive effect on lipid metabolism, stimulating mineral absorption, and immunomodulatory properties [11,18] (Table 2).

Why are some oral nutritional supplements added with functional oligosaccharides/prebiotic fibers such as GOSs, FOSs, or a mixture? See Figure 2 [19,20].

#### 3.1.1. Fructo-Oligosaccharides (FOSs)

Fructo-oligosaccharides and inulin (fructans) are made up of fructose chains of variable lengths with a terminal glucose molecule. Fructose units are joined by β (1→2) bonds while glucose units are connected by α (1→1) bonds, similar to that present in sucrose. From a chemical point of view, FOSs and inulin differ in their degree of polymerization (DP): from 3 to 10 for FOSs, up to 60 for inulin (polymer). Fructo-oligosaccharides (FOSs) remain structurally unchanged as they bypass hydrolysis by small intestinal glycosidases and reach the cecum. In the intestinal environment, they undergo metabolism by the gut microflora, resulting in the production of short-chain carboxylic acids, L-lactate, CO(2), hydrogen, and other metabolites. FOSs possess several intriguing characteristics, including mild sweetness, zero calorie content, non-cariogenic properties, and recognition as soluble dietary fiber. Additionally, FOSs offer significant physiological benefits, such as reduced carcinogenicity, a prebiotic impact, enhanced mineral absorption, and lowered levels of serum cholesterol, triacylglycerols, and phospholipids [21].

Although consensus on the optimal type and quantity of fiber for children remains elusive, evidence suggests that dietary fibers play a crucial role in maintaining healthy gastrointestinal functions and preventing childhood constipation. Constipation, a prevalent condition affecting 3–29% of children globally, often begins in infancy and persists in 35–52% of cases for years [22]. The ESPGHAN/NAPGHAN guidelines advocate for a standard dietary fiber intake for children with functional constipation, although data on the most advantageous fiber type or source for these children are scarce, and recommendations are lacking [23]. Recently, a prospective study tested a mixture of polydextrose (4.17 g) and FOSs (0.45 g) in a daily dose of food supplement in 77 children. The number of children with less than three bowel movements per week dropped from 59.7% to 11.7%, hard stools (Bristol type 1 and 2) decreased from 68.8% to 7.8%, pain during defecation decreased from 79.2% to 10.4%, fear of defecation decreased from 68.8% to 3.9%, and the number of children with abdominal pain symptoms reduced from 84.2% to 2.6% at the end of the study. Significantly, there was no comparison conducted with a control group [24].

Microbiome-targeted treatments, such as fiber and prebiotics, hold promise for rebalancing the gut microbiome and improving both gut health and clinical outcomes for the host. Indeed, research on adult patients with inflammatory bowel disease (IBD) shows that fiber and prebiotics can positively influence the microbiome and ameliorate disease progression. However, as of now, there have been no studies assessing the therapeutic efficacy of fiber and prebiotics in pediatric IBD patients [25].

#### 3.1.2. Biological Effects of Galacto-Oligosaccharides (GOSs)

This paper tries to also explore the association of galacto-oligosaccharides with oral nutritional supplements, aiming to understand their potential health-promoting properties. These investigations explore the impact of such supplements on various aspects of health, including intestinal barrier function, lactose intolerance, constipation relief, and potentially other health-related outcomes. Researchers have aimed to understand the potential advantages and mechanisms of action associated with GOSs in oral nutritional supplements. GOSs have been shown to enhance the production of health-related short-chain fatty acids (SCFAs), stimulate the growth and differentiation of colonic epithelial cells, improve energy transduction in colonocytes, and influence lipid and carbohydrate metabolism. GOSs serve essential functions in milk and are a widely employed prebiotic substance [11]. The presence of GOSs in the diet has been associated with various beneficial biological activities. Research conducted by He et al. in 2021 indicated that the supplementation of GOSs to a high-fat diet effectively stimulates the growth of bifidobacteria. Bifidobacteria, known for their ability to produce lactic acid, contribute to the acidification of the intestinal environment [26]. This acidification helps in restraining the growth of pathogenic bacteria and enhances the mucosal barrier’s function. Infants who received prebiotic supplements had softer stools compared to those receiving standard formula, with stool consistency resembling that of breastfed infants.

Numerous studies in both animals and humans have indicated the positive impact of GOSs on bone composition and structure. Several mechanisms have been proposed: firstly, bacterial fermentation of acidic metabolites in the colon lowers the local pH in the intestine, leading to an increased luminal concentration of calcium ions and enhanced passive calcium absorption; secondly, short-chain fatty acids (SCFAs) alter the charge of calcium, stimulate calcium channels, and promote increased calcium absorption [27].

GOSs perform a crucial function in alleviating lactose intolerance and preventing constipation (Figure 3 [28]). Oligosaccharides found in human milk contribute to the development of beneficial intestinal microbiota in infants, particularly *Bifidobacterium* and *Lactobacillus* [29]. While human milk oligosaccharides are limited and primarily found in the colostrum of various mammals, making the cost of HMOs high, GOSs derived from lactose exhibits notable prebiotic properties and share a structural resemblance to HMOs. Comprising two to seven galactose units with a glucose molecule at the terminus, GOSs are a cost-effective alternative [28]. Oral nutritional supplements enriched with GOSs present a promising avenue for promoting health and addressing specific gastrointestinal issues. This review consolidates existing knowledge, highlighting the potential benefits of incorporating GOSs into oral nutritional supplements and suggesting avenues for future research.

## 4. Oral Nutritional Supplements Enriched with Transforming Growth Factor-Beta 2 (TGF-Beta)

Nutritional support plays an important role in the treatment of inflammatory bowel diseases (IBDs). Oral nutritional supplements enriched with specific bioactive peptides seem to have a direct effect on the intestinal mucosa by suppressing the inflammatory process [30]. Moreover, supplementary enteral nutrition after primary therapy and after remission is induced may be associated with the prolongation of remission and promotion of linear growth in patients with IBD [31], particularly in Crohn’s disease (CD), a chronic, relapsing transmural inflammation characterized by skip intestinal lesions anywhere in the GI tract [32]. The leading biocompound that appears to play important roles in the inflammatory processes exhibited in IBD is the transforming growth factor (TGF)-β superfamily (thirty-three genes that encode for homodimeric or heterodimeric secreted cytokines) [33], which includes TGF-β (TGF-β1, TGF-β2, and TGF-β3) and related proteins (activins, growth and differentiation factors (GDFs), the bone morphogenetic proteins (BMPs), the müllerian inhibiting substance (MIS), and the nodal). TGF-β is a multifunctional cytokine expressed by many types of cells [34], such as epithelial cells, fibroblasts, and immune cells, particularly leukocytes and macrophages, with various immunomodulatory roles especially relevant to the GI tract. This cytokine contributes to the regulation of several key cellular functions, including cell proliferation, differentiation, migration, apoptosis, and extracellular matrix production [35]. TGF-β ligands bind to pairs of transmembrane receptors known as receptor types I and II [36]. The resulting TGF-β receptor complex triggers intracellular signaling via both the Smad-dependent canonical and the Smad-independent non-canonical pathways. SMAD proteins are the major effector molecules in the TGF-β signaling pathway, through phosphorylation at specific Ser residues in their C-terminal regions. These SMAD proteins, called “receptor-regulated SMADs” (R-SMADs), associate with the common mediator SMAD4 protein, form trimeric complexes, and are then shuttled to the nucleus. These complexes can directly bind on specific DNA sequences that have been characterized as being SMAD-binding elements (SBEs) and cooperate with DNA-binding transcription factors and chromatin modifiers to regulate the expression of TGF-β-responsive genes (Figure 4a,b) [37].

SMADs are capable of regulating gene expression at the post-transcriptional level as well, impacting mRNA splicing, stability, and translation through interactions with RNA-binding proteins (RBPs) and non-coding RNAs (ncRNAs) [38]. Among the various recognized mechanisms, TGF-β seems to be a key modulator of innate and adaptive immunity, acting as a general enforcer of immune tolerance and a suppressor of inflammation. For this reason, TGF-β has been considered, in particular, for the treatment of IBD, administered directly or through formulas added with TGF-β content. Some studies have shown that the consumption of a formula enriched with TGF-β is associated with a reduction in the expression of cytokines in the intestinal mucosa with a concomitant improvement in histologic parameters in children and teenagers [39]. TGF-β regulates multiple immune processes of T-cells. A major function of TGF-β signaling in T-cells is to regulate and balance the differentiation of naive T-cells into specific effector subsets, in particular, to suppress T-cell proliferation and activation through Treg differentiation. TGF-β inhibits, through the induction of Foxp3 (a master regulator of Tregs), the maturation of naive CD4+ T-cells into TH1 and TH2 T helper cells and of naive CD8+ T-cells into cytotoxic T lymphocytes (CTLs), promoting instead their differentiation to the Treg stage [40], a specialized subpopulation of T-cells that act to suppress the immune response, is able to inhibit T-cell proliferation and cytokine production, and plays a critical role in preventing autoimmunity.

TGF-β also inhibits the function of dendritic cells (DCs), responsible for the antigen presentation to CD4+ T-cells and CD8+ T-cells [41] and suppresses natural killer (NK) cells and their production of interferon-γ (IFN-γ) (Figure 5) [42].

TGF-β also plays an integral role in the process of healing of the intestinal mucosa and in the development of fibrosis and strictures [43] through regulation of fibroblast activity, the main producers of connective tissue matrix which play a key role in tissue repair. TGF-β potently induces the recruitment, proliferation, and activation of fibroblasts that produce collagens, fibronectin, and other components required for extracellular matrix (ECM) assembly, as well as integrins that mediate cell adhesion to the ECM [44] (see Table 3).

In studies conducted on nutritional supports enriched with TGF-β, was found an improvement on mucosal healing and remission induction. Furthermore, it was demonstrated that C-Reactive Protein levels, sedimentation rates, reductions in PCDAI, and albumin levels showed significant improvement in these patients [45,46]. In the light of these data, oral nutritional supplements enriched with TGF-β, given in addition to medical treatments without restricting normal diets, avoided the potential side effects of steroids in CD patients, remained in remission for longer durations, and contributed to improving anthropometrics data.

In studies investigating oral nutritional supplements enhanced with TGF-beta (transforming growth factor-beta), no serious side effects were reported among the patients. The minor side effects that were occasionally observed, such as abdominal pain and abdominal distention, were infrequent and typically resolved quickly [47].

## 5. Immunonutrition and Oral Nutritional Supplements

Immunonutrition (IN) is characterized as the utilization of particular nutritional substrates, referred to as “immunonutrients”, which possess the capability to modulate specific mechanisms involved in various immune and inflammatory pathways. Numerous nutrients could fit into this description, and among these, the most defined are Omega 3 polyunsaturated fatty acids, arginine, branched-chain amino acids, trace metals (e.g., zinc, copper, and iron), vitamin D, and nucleotides [48]. These immunonutrients primarily target mucosal barrier function, cellular defense, and either local or systemic inflammation [49]. Recent research works have indicated that combinations of immunonutrients, incorporating omega-3 fatty acids, glutamine, arginine, and nucleotides, may be advantageous for certain patient populations. This is usually associated with a decrease in both hospital length of stay and infection rates. Nevertheless, there was no observed reduction in mortality. Moreover, various immunonutrients like proteins, vitamins, trace metals, and enzymes possess antioxidant properties, mitigating tissue damage and lowering the risk of carcinogenesis [50]. Therefore, the proposition of utilizing oral nutritional supplements enriched with certain immunonutrients is suggested as a nutritional treatment approach. This method could serve as an option to meet nutritional needs and regulate the immune response in patients with particular pathologies, cancer, or those preparing for surgery. However, evidence-based recommendations for the use of these formulations in pediatric clinical practice are scarce and confined to specific populations. This discussion focuses on the characteristic changes in innate and acquired immunity that occur during critical illness, as well as the mechanisms through which immune nutrients can beneficially influence the immune response (Table 4).

### 5.1. Arginine

L-arginine is categorized as a conditionally essential amino acid. In fact, there are instances, such as developmental stages and certain pathological conditions (infection or inflammation, or under conditions in which renal and/or intestinal metabolic functions are impaired), where the endogenous production is insufficient. Consequently, a dietary intake becomes necessary in these situations [51]. Arginine freely available in the body is sourced from the diet (it is abundant in seafood, watermelon juice, nuts, seeds, algae, meats, rice protein concentrate, and soy protein isolate [52]), de novo synthesis, and protein turnover. While the synthesis of arginine from citrulline is possible in various cell types [53], a significant portion of endogenous synthesis takes place through a cooperative process involving the epithelial cells of the small intestine and the proximal tubule cells of the kidney [54], a pathway known as the “Intestinal-Renal Axis” of arginine synthesis. In fact, the gut is the main source of citrulline for the synthesis of L-arginine. Enterocytes in the small intestine express carbamyl phosphate synthetase I and ornithine transcarbamylase. This expression allows for the synthesis of citrulline from glutamine, proline, or ornithine. Citrulline is absorbed by the proximal tubules of the kidney and undergoes efficient conversion to arginine through the sequential action of argininosuccinate synthetase and argininosuccinate lyase (ASL) (Figure 6) [54].

Within the liver, possessing a fully functional urea cycle, arginine is synthesized from ornithine, carbamoyl phosphate, and the amino group sourced from aspartate. However, the hepatic urea cycle does not yield a net production of arginine, as the synthesis of arginine within the cycle precisely matches its breakdown. Consequently, the liver does not serve as a net source of arginine.

Arginine serves as the precursor for several crucial substances within the organism, including proteins, nitric oxide, proline, creatine, agmatine, and polyamine. Additionally, it can trigger the secretion of hormones such as insulin, glucagon, growth hormone, and prolactin. Furthermore, it plays a role in immunoregulation [55]. Numerous enzymes are involved in the degradation of arginine, including arginase, three variants of nitric oxide synthase (NOS), arginine decarboxylase (ADC), and arginine:glycine amidinotransferase (AGAT) [56]. The three NOS isoenzymes (I/nNOS; II/iNOS; and III/eNOS) [57] exhibit variations in structural properties, distribution, regulation, and the production of nitric oxide quantities. Each of the three NOS enzymes utilizes L-arginine as a substrate, leading to a reaction where N-hydroxy-L-arginine (L-NOHA) serves as an intermediate, releasing both nitric oxide (NO) and L-citrulline [58]. NOS I and NOS III operate in a Ca^2+^-dependent and constitutively expressed manner, while NOS II functions independently of Ca^2+^ and is abundantly expressed in response to immunological challenges [56]. NOS relies on various cofactors, such as flavin adeninedinucleotide (FAD), flavin mononucleotide (FMN), heme, and tetrahydrobiopterin (BH4), which is crucial and acts as the rate-limiting factor [59]. NO governs multiple signaling pathways across various tissues, playing diverse physiological roles. Its most recognized and established functions are associated with the immune, cardiovascular, and neuronal systems. Notably, NO is generated by various immune cells, primarily macrophages, serving as a crucial regulator of immunity and inflammation [60]. Within the cardiovascular system, NO derived from the endothelium acts as a potent vasodilator, playing a pivotal role in determining vascular tone and blood pressure [61]. Finally, in the nervous system, NO originating from neurons is recognized for its role in regulating neural development and influencing diverse brain functions, including cognition and response to stress [62]. New findings indicate that NO plays a significant role in governing the homeostasis of both the fetus and neonate [63].

Similar to the production of NO, arginine plays a unique role as the exclusive amino acid supplying the amidino group for creatine synthesis. In this process, arginine acts as a donor for transferring the amidino group to a glycine backbone. The catalysis of this reaction is carried out by L-arginine glycine amidinotransferase (AGAT), resulting in the generation of ornithine and guanidinoacetic acid. Creatine plays a major role in energy metabolism in skeletal muscle and neuronal cells [64]. Instead, arginase is a classical enzyme in the urea cycle that facilitates the hydrolysis of L-arginine into urea and L-ornithine. There are two distinct forms of mammalian arginase: type I and type II, each governed by separate genes. Type I arginase is a cytosolic enzyme prominently present in the liver. It plays a crucial role as a key element in the transport, storage, and excretion of nitrogen, serving as an essential factor in ammonia detoxification through the urea cycle. This function aids in averting metabolic disruptions caused by heightened levels of tissue ammonia [65]. Type II arginase is a mitochondrial enzyme characterized by modest expression levels in various tissues, such as the small intestine, kidney, brain, endothelium, mammary gland, and macrophages [66]. It plays an important role in regulating the synthesis of NO, proline, and polyamines [67]. Polyamines, such as putrescine, spermine, and spermidine, play roles in membrane transport as well as in cell growth, cell proliferation, and cell differentiation [68]. Additionally, arginine triggers the release of growth hormones and insulin in mammals, including preterm infants, thereby playing a crucial role in overseeing protein, lipid, and carbohydrate metabolism [69].

The significance of arginine in metabolic pathways leading to protein synthesis and the elimination of nitrogenous waste, along with its involvement in cell signaling, proliferation, differentiation, and stimulation of the immune system, is firmly established. What still needs clarification is whether dietary arginine can be supplemented at pharmacological levels to influence metabolic outcomes. In fact, maintaining an adequate supply of arginine has been closely linked to enhancing immune responses. Beyond its role as a building block for protein synthesis, arginine functions as a substrate for various metabolic pathways that significantly impact the biology of immune cells, particularly macrophages, dendritic cells, and T-cells. For instance, the administration of additional arginine exhibits positive effects on the immune system, specifically influencing thymus-dependent and T-cell-dependent immune responses. Initial animal experiments illustrated the thymotropic effects of arginine, evident in increased thymic weight, elevated thymic lymphocyte content, and heightened reactivity of thymic lymphocytes [65]. In various experimental models, additional immunomodulatory effects of arginine have been observed. Arginine supplementation has been shown to alleviate the diminished delayed-type hypersensitivity response typically associated with extreme youth age [70]. In 2001, Waugh and colleagues conducted a preliminary phase II clinical trial involving five pediatric participants (aged 10–18 years) diagnosed with sickle cell disease. Their trial revealed that administering oral L-citrulline at doses ranging from 0.09 to 0.13 g/kg twice daily over a four-week period led to a 65% increase in plasma arginine concentrations, resulting in the normalization of leukocyte and neutrophil counts [71].

In a double-blind, randomized, placebo-controlled trial investigating enteral arginine supplementation in burned children, Veronica B. Marin et al. aimed to address the existing gap in data for pediatric burn patients [72]. Their study compared the effects of supplemental dietary arginine with an isocaloric and isonitrogenous placebo diet, seeking to elucidate the specific roles of arginine in immune and metabolic responses in children affected by burns. Twenty-eight children who met the criteria were randomly divided into two groups: one receiving an arginine-supplemented diet (AG; *n* = 14) and the other receiving an isocaloric isonitrogenous diet (CG; control, *n* = 14) for a duration of 14 days. This study found that a diet exclusively supplemented with arginine enhances mitogen-stimulated lymphocyte proliferation in burned children. The immunological advantage noted in the AG group was linked to heightened arginine disposal and metabolism. This was evidenced by elevated ornithine levels resulting from the pharmacokinetic properties of arginine, including increased turnover and oxidation, restricted synthesis, and irreversible conversion to ornithine. Following the noted immunostimulatory effects of arginine, there has been a growing interest in dietary arginine supplementation, aiming to formulate what are referred to as immune-enhancing diets [73]. Nutrition formulas enriched with arginine have demonstrated benefits in both animal and human trials. However, there is limited evidence suggesting that arginine alone is accountable for these positive effects, as the immune-enhancing diets also included other pharmacologically active components [74].

Reports of side effects from arginine administration are primarily derived from animal studies or in vitro studies, which indicate that these side effects are minimal and generally transient. Theoretically, an overdose of arginine could lead to hypotension due to its vasodilatory effects [75].

### 5.2. Omega 3

Omega-3 (*n*-3) are polyunsaturated fatty acids (PUFAs) characterized by a double bond at the *n*-3 position. These PUFAs include eicosapentaenoic acid (EPA), alpha-linolenic acid (ALA), and docosahexaenoic acid (DHA). Mammals lack the ability to produce the essential alpha-linolenic acid (ALA). Nevertheless, they have the capacity to synthesize EPA and DHA from alpha-linolenic acid, albeit at a conversion rate of less than 5% for EPA and 1% for DHA. As a result, mammals rely on acquiring these fatty acids from their diet. EPA and DHA predominantly come from the marine ecosystem, with fish and seafood serving as the primary reservoir of *n*-3 PUFAs [76]. Additionally, leafy vegetables and nuts are renowned for their elevated content of *n*-3 FAs, establishing themselves as the principal source of ALA [77].

Numerous studies have identified the potential health advantages associated with the intake of *n*-3 PUFAs, such as changes in the physical characteristics of the membrane (referred to as “fluidity”) [78], impacts on cellular signaling pathways [79], or changes in the production pattern of lipid mediators [80].

Publications from the early 1990s on human infants indicated that preterm infants, when fed a formula enriched with *n*-3 LCPUFA, primarily in the form of DHA, exhibited enhanced retinal sensitivity and visual acuity in comparison to preterm infants fed the standard unsupplemented formulas of that time [81,82]. Research has also brought attention to the potential impact of extremely long-chain omega-3 fatty acids on mental development, enhancing childhood learning and behavior [83]. For instance, research has shown a direct correlation between elevated levels of DHA in the mother’s plasma, especially in breast milk, and enhanced growth and development of the brain and visual system in children [84]. However, these areas of potential action are still subjects of controversy and demand more substantial scientific support. On this matter, there is a hypothesis suggesting that *n*-3 PUFAs play a role in influencing the structure and function of biological membranes, such as elasticity, membrane organization, and ion permeability. Consequently, these fatty acids may potentially aid in brain glucose uptake, neurotransmission, and neuronal function [85].

Many studies have examined the various mechanisms through which omega-3 fatty acids exert their beneficial actions on the body. In one study, researchers demonstrated that DHA induced PPARγ (peroxisome proliferator-activated receptor gamma) and several established target genes of PPARγ in dendritic cells. These outcomes were associated with a reduction in the production of the inflammatory cytokines TNFα and IL-6 following endotoxin stimulation. Consequently, by activating PPARs, *n*-3 PUFAs can govern metabolism and various cellular and tissue responses, such as adipocyte differentiation and inflammation [86,87]. Moreover, EPA and DHA exhibit the capability to hinder the production of various inflammatory proteins, including COX-2, inducible NO synthase, TNFα, IL-1, IL-6, IL-8, and IL-12 in cultured endothelial cells [88], monocytes [89], macrophages [90], and dendritic cells [91]. These inhibitory effects of *n*-3 PUFAs appear to be associated with reduced IkB phosphorylation and diminished activation of NFκB [92], a crucial transcription factor that triggers the expression of genes encoding diverse proteins linked to inflammation and necessitates the activation achieved through phosphorylation of its inhibitory subunit, IkB. Additionally, *n*-3 PUFAs reduce the accessibility of arachidonic acid for eicosanoid synthesis, impeding its metabolism. This reduction in eicosanoid production from arachidonic acid can impact the actions regulated by these mediators [93].

Interventional trials with *n*-3 PUFAs have not reported any serious therapy-related side effects. Compared to the placebo, *n*-3 PUFAs show similar rates of adverse reactions during short-term use, primarily including nausea and other gastrointestinal symptoms [94].

### 5.3. Nucleotides

Nucleotides (NTs) play a crucial role in nearly all biological processes within the body, serving as essential components of nucleic acids, namely DNA and RNA. NTs are composed of heterocyclic nitrogenous bases, which can be either pyrimidine or purine. Nucleotides can be acquired either through the presence of nucleoproteins (proteins linked to a nucleic acid) naturally found in all foods of both animal and plant origin (healthy individuals typically consume 1–2 g of nucleotides daily through their diet [95]) or through endogenous synthesis, which serves as the primary source of nucleotides [96]. Due to the substantial presence of NTs in breast milk, they are regarded as conditionally essential nutrients for infants. These biocompounds hold significant physiological significance in supporting immune function, lipid metabolism, and growth. Primarily, they act as precursors to nucleic acids—monomeric units found in DNA and RNA, pivotal in storing and transferring genetic information, facilitating cell division, and enabling protein synthesis [97]. Moreover, nucleotides and their derivatives play various roles in energy metabolism (adenosine triphosphate—ATP), enzymatic regulation, and signal transduction and serve as structural components of coenzymes such as NAD, coenzyme A, and NADP+ [98]. Recent research on the impact of NTs on the immune system suggests their potential as immunomodulators. In vitro and animal model studies indicate that nucleotides stimulate the differentiation and proliferation of lymphocytes. Consequently, stages of lymphocyte activation and function also affect nucleotide metabolism to some extent, leading to increase de novo synthesis and salvage in stimulated lymphocytes [99]. Furthermore, the absence of nucleotides led to substantial reductions in host immune responses, resulting in the downregulation of T-cell function and antigen stimulation [100]. Supplementing the diet with nucleotides improves disease resistance, potentially ascribed to increased peritoneal macrophage phagocytosis, heightened NK cell activity, augmented secretion of T-cell-dependent antibodies, elevated production of interleukin-2, and increased levels of bone marrow cells and neutrophils in peripheral blood. Moreover, NT supplementation has the potential to enhance growth, immunity, and stress tolerance, modify intestinal structure, and strengthen the body’s defenses against viral, bacterial, and parasitic infections. Nucleotides have also exhibited robust anti-inflammatory properties. Specifically, in vitro studies involving macrophages, or the endothelium have demonstrated that extracellular adenosine can effectively downregulate the potent inflammatory cytokine, tumor necrosis factor alpha (TNF-α) [101]. As crucial elements in immunonutrition, nucleotides showcase diverse physiological activities. These include improving the prognosis of individuals undergoing physiological stress, reducing levels of DNA damage, extending lifespan, serving as anti-fatigue agents, and accelerating carbon turnover in the fundic stomach. Research has suggested that dietary nucleotides play a vital role in intestinal development, liver and immune function, and the rapid growth of cells under physiological stress [102].

To investigate the impact of nucleotides on immune function and sepsis in infants, two clinical studies were undertaken involving healthy term young children. These infants were either breastfed (HM group) or fed with one of two infant formulas—one supplemented with nucleotides (NFM group) and the other not supplemented (FM group). Physical growth, hematological indices, and plasma biochemistry profiles were not significantly different among the three groups. Natural killer cell activity and interleukin-2 activity were significantly higher at 2 months of age in the NFM group compared to the FM group. However, no significant difference was found at 4 months of age [103]. Additional research exploring the potential advantageous impact of dietary nucleotides, in particular clinical scenarios among high-risk neonates, is recommended. These scenarios encompass extreme prematurity, small for gestational age (SGA), gut injuries such as necrotizing enterocolitis (NEC), and extended periods of restricted enteral or parenteral nutrient intake.

## 6. Potential Bioactive Candidates Added to ONSs

Bioactive peptides exhibit diverse biological actions influenced by factors such as amino acid class, net charge, secondary structures, sequence, and molecular mass. Numerous studies have identified the bioactivities of peptides, associating them with enhanced overall health and reduced risk of specific chronic diseases, including cancer, diabetes, and heart diseases (Figure 7) [104].

### 6.1. Glutamine

Glutamine is the most abundant and versatile amino acid in the body, constituting greater than 50% of the total free amino acid pool [105], with a concentration of between 0.6 and 0.7 mmol/L. This molecule takes on the role of a conditionally essential amino acid in situations where there is a deficiency, such as in sepsis [106], traumas [107], cancer [108], infections [109], surgeries [110], or prolonged intense physical exercise [111]. In a study examining glutamine levels in critically ill children, it was observed that they experienced significant early glutamine depletion (glutamine 0.31 mmol/L) compared to convalescent levels (0.40 mmol/L). The glutamine levels during the acute phase of illness were 52% below the lower limit of the normal reference range, while in the convalescent samples, levels were 26% below, suggesting depletion rather than an ongoing consequence of fluid shifts [112]. Nevertheless, the utilization of glutamine in children is not clearly defined, and there is a lack of sufficient data regarding the advantages of glutamine as a pharmacological agent in pediatric critical illness. Anyway, a reduction in plasma glutamine availability has been noted to contribute to compromised immune function in various clinical conditions. Specifically, glutamine depletion diminishes lymphocyte proliferation, hinders the expression of surface activation proteins, reduces cytokine production, and triggers apoptosis in these cells [113]. Present in nearly every cell, glutamine serves as a substrate for various biosynthetic pathways essential for cellular integrity and function, including nucleotide synthesis (purines, pyrimidines, and amino sugars), nicotinamide adenine dinucleotide phosphate (NADPH), antioxidants, and more [114]. The concentration and availability of glutamine in the entire body hinge on the equilibrium between its synthesis and/or release and its uptake by human organs and tissues.

The two main intracellular enzymes involved in glutamine metabolism are glutamine synthetase (GS), found in the cytosol, which initiates the reaction that synthesizes glutamine from ammonium ions (NH^4+^) and glutamate, involving the consumption of ATP, and phosphate-dependent glutaminase (GLS), mainly found within the mitochondria, which is accountable for the hydrolysis of glutamine, transforming it back into glutamate and NH^4+^ (Figure 8).

Mitochondrial glutamate is transformed into alpha-ketoglutarate (α-KG) through the action of glutamate dehydrogenase 1 (GLUD1 or GDH1) or various mitochondrial aminotransferases. α-KG can be exported from the mitochondria to the cytosol through SLC25A11, where it engages in fatty acid biosynthesis and NADH generation, or can then participate in the tricarboxylic acid (TCA) cycle, supporting either the oxidative phosphorylation (OXPHOS) pathway or the reductive carboxylation pathway [115,116]. The primary contributors of glutamine, with notable activity in tissue-specific glutamine synthesis (GS), include the lungs, liver, brain, adipose tissue, and skeletal muscles. Conversely, through the actions of enzymes like GLS that degrade glutamine, the amino acid can undergo degradation at varying rates. Tissues such as the intestinal mucosa, renal tubules, and, particularly, leukocytes are prominent consumers of glutamine [117]. In fact, glutamine serves as a vital energy source for leukocytes, enough to be considered “fuel for the immune system” [118]. Lymphocytes and macrophages have high rates of glutamine utilization, especially during inflammatory states such as sepsis and injury. In a study, Newsholme and Parry-Billings [119] established a strong correlation between the phagocytosis rate in murine macrophages and the concentrations of glutamine. Furthermore, in vitro studies have shown that glutamine supplementation in vitro optimizes macrophage functions [118]. On the other hand, a reduction in plasma glutamine availability has been documented as a factor contributing to compromised immune function in various clinical scenarios. Specifically, the depletion of glutamine diminishes the proliferation of lymphocytes, hinders the expression of surface activation proteins, reduces cytokine production, and triggers apoptosis in these cells [113]. The role of glutamine extends to its significant antioxidant capabilities, particularly crucial in the context of critical and severe illnesses. In fact, glutamine acts as a precursor for the formation of glutathione (GSH) through the conversion of glutamate. In rat-based in vivo experiments, it has been shown that administering glutamine before ischemia/reperfusion injury or surgical procedures can boost GSH concentrations [120], enhancing cellular resilience against damage and mitigating oxidative stress [118]. Another noteworthy aspect is the function of glutamine as the favored energy source for intestinal epithelial cells [121]. Remarkably, glutamine stands out as the most plentiful free amino acid found in milk. This observation implies that glutamine could play a role in supporting intestinal health and development during the lactation period [122]. It is well known that the small intestinal epithelium undergoes frequent renewal, experiencing regeneration approximately every 2–5 days, a continuously high level of cell proliferation required to maintain homeostasis [123]. Glutamine exerts its influence on various signaling pathways (Figure 9) [121] responsible for regulating the cell cycle and proliferation, such as epidermal growth factor (EGF), transforming growth factor, mitogen-activated protein kinases (MAPKs), and insulin-like growth factor 1 (IGF-1) [124,125]. Various sources of evidence have suggested that glutamine also regulates the expression of tight junction proteins (occludin, the claudin-family proteins, junction adhesion molecules (JAMs), zonula occludens (ZO-1, ZO-2, and ZO-3), which create a physical barrier that seals neighboring epithelial cells, establishing a separation between epithelial and endothelial cells [126]. The reduction of glutamine is correlated with a decrease in the tight junction protein claudin-1, resulting in the breakdown of barrier function [127]. This breakdown is often associated with the initiation of various gastrointestinal diseases, which are frequently linked to the initial disruption of tight junctions. Other significant factors contributing to the development of intestinal conditions like ulcerative colitis, Crohn’s disease, and colorectal cancer include inflammatory processes [128]. Multiple lines of evidence indicate that glutamine demonstrates anti-inflammatory properties by modulating various inflammatory signaling pathways, including those involving nuclear factor κB (NF-κB) (an essential controller of immune responses with implications in the pathogenesis of various inflammatory diseases) and signal transducer and activator of transcription (STAT) [129]. STAT proteins serve as transcription factors regulating the immune system, cellular proliferation, and development [130]. Glutamine plays a role in inhibiting STAT activation and suppressing the expression of inflammatory cytokines, such as IL-6 and IL-8, in intestinal tissues.

There is also a strong association between intestinal inflammatory disorders and increased apoptosis of intestinal epithelial cells. Glutamine has been demonstrated to exhibit anti-apoptotic properties in the intestine through various mechanisms: (1) by maintaining normal cellular redox status as a precursor for GSH [131]; (2) by regulation of caspase activation, a group of protease enzymes that holds significant roles in triggering apoptosis [132]; and (3) enhancing the expression of heat shock proteins (HSPs) (i.e., HSP10, HSP40, HSP60, HSP70, HSP90 and HSP100), proteins with chaperone function, preventing the aggregation of newly synthesized polypeptide chains during folding or by clearing improperly folded and unfolded proteins [133], through glutamine-mediated increase the heat shock factors (HSF1) expression, activator of HSPs [134].

In conclusion, the dietary presence of glutamine has been demonstrated to play a crucial role in maintaining the integrity of the intestinal mucosal barrier. This is achieved through the regulation of gene and protein expression related to cell proliferation, differentiation, apoptosis, protein turnover, anti-oxidative properties, and immune responses. For this reason, a new ONS added with glutamine should be evaluated in the future, involving in vivo studies to enhance our understanding of the molecular mechanisms underlying the impact of glutamine on gut barrier function and to contribute to the development of personalized glutamine regimens for the therapeutic management of various diseases.

### 6.2. Lactoferrin

Lactoferrin is a non-heme iron-binding glycoprotein found in exocrine biological fluids like breast milk, tears, bronchial secretions, and gastrointestinal fluids, playing a crucial role in both human and bovine milk [135]. Lactoferrin, possessing antimicrobial and immune-modulating properties, is released from activated neutrophil granules, leading to increased concentrations in plasma and feces during infection and inflammation due to neutrophil recruitment. Human lactoferrin (hLf) shares structural and functional similarities with bovine lactoferrin (bLf). Although lactoferrin concentrations are notably higher in human milk compared to bovine milk, bLf can be efficiently extracted from bovine milk in substantial amounts. Bovine lactoferrin is progressively utilized as a nutritional supplement across various patient populations and conditions and is generally considered safe and well-tolerated.

Specifically, given that bLf receptors are present in the intestinal mucosa and lymphatic tissue cells of the intestine, the preservation of bLf’s structural integrity is crucial for binding to these receptors [136]. Nevertheless, studies have demonstrated that bLf can directly stimulate the growth and proliferation of enterocytes, depending on its concentration [137]. Consequently, the absorption of lactoferrin in the intestine may vary during different life stages. During early life, infants who are breastfed or fed with infant formula fortified with bLf will experience a high concentration of lactoferrin in the intestinal lumen. This is attributed to minimal proteolytic degradation and heightened cell proliferation.

The mucosal development stimulated by lactoferrin can lead to an increase in mucosal surface area, thereby improving the absorption not only of iron but also of other nutrients. As the infant matures, protein digestion becomes more efficient, resulting in a significant decrease in lactoferrin concentration and an increase in differentiation.

To ensure the orally administered protein reaches the intestine without undergoing degradation in the stomach, protective measures must be implemented. Various approaches have been employed in formulating bLf oral delivery systems to enhance its oral bioavailability. Some of the commonly utilized methods to safeguard bLf during the oral and gastric passage include iron saturation, microencapsulation, PEGylation, and absorption enhancers [138].

The evidence supporting the impact of lactoferrin supplementation on inflammation, immune function, and respiratory tract infections (RTIs) in humans has demonstrated the crucial role in host defense by performing a range of physiological functions, encompassing antiviral, antimicrobial, antioxidant, and immunomodulatory activities. The protective functions of lactoferrin can rely on its capacity to bind iron or operate independently of it [139]. Probably, as Kruzel et al. [140] reviewed, through iron sequestration, LTF regulates the natural equilibrium of reactive oxygen species (ROS) production and their elimination rate, thereby providing a natural defense against direct oxidative cell damage. A compelling hypothesis suggests that by managing oxidative stress, LTF influences the responsiveness of the innate immune system, leading to changes in the production of immune regulatory mediators crucial for shaping the development of adaptive immune function. Numerous studies have indeed demonstrated the significant modulatory effects of LTF on the adaptive immune system.

In recent times, it has been that lactoferrin may play a role in safeguarding against common viral infections by boosting type I interferon (IFN) production, enhancing NK cell activity, and promoting type 1 T-helper (Th1) cell cytokine responses [141]. Numerous in vitro studies have indicated that lactoferrin treatment hinders virus growth, cellular entry, and the cytopathic effect after exposure to prevalent respiratory viruses, such as respiratory syncytial virus [142] and influenza virus [143]. Furthermore, colostrum, rich in lactoferrin at approximately 7 g/L, has demonstrated protective effects against RTIs in both adults and children [144]. Recent preliminary evidence using pseudo-viruses has suggested that Lf may provide protection against emerging viruses like severe acute respiratory syndrome coronavirus 2 (SARS-CoV-2), which was responsible for the COVID-19 pandemic [145].

Many points of evidence from several randomized clinical trials support the role of bLf in reducing systemic inflammation, specifically IL-6, due to various mechanisms. The regulation of iron homeostasis by bLf is emerging as the most pertinent aspect in the context of the presented evidence; in particular, IL-6 downregulates ferroportin, the transporter responsible for moving iron from tissues to the systemic circulation, while IL-6 upregulates hepcidin, which inhibits iron transport [146].

As the RCT from Widjaja et al. [147] showed, the lactoferrin in oral nutritional supplements affects IL-6 and IL-10 levels in children experiencing failure to thrive and infections over a 90-day intervention, functions as a regulator of innate immunity and serves as a defense mechanism owing to its antimicrobial properties.

Lactoferrin possesses both bacteriostatic and bactericidal effects while also being capable of stimulating the immune response in an organism. This helps mitigate tissue damage resulting from an excessive pro-inflammatory response, often seen in chronic infections. Consequently, it has the potential to decrease the occurrence of acute gastrointestinal symptoms and shorten the duration of respiratory symptoms in children under 12 months old, whether stemming from viral or bacterial infections.

In a 2020 randomized, double-blind, placebo-controlled trial, researchers investigated the effects of lactoferrin (LF)-fortified formula on acute gastrointestinal and respiratory symptoms in children [148]. As result, the occurrence of acute gastrointestinal symptoms was notably lower in the LF group (22 out of 53 (41.5%)) compared to the placebo group (30 out of 48 (62.5%), *p* = 0.046). Additionally, the overall duration of acute respiratory symptoms was significantly reduced in the LF group (9.0 days) compared to the placebo group (15.0 days, *p* = 0.030).

It is becoming evident that LTF plays a pivotal role in shaping and fully activating adaptive host immune responses. At the molecular level, homeostasis is governed by a complex network involving the neuroendocrine and immune systems, where LTF assumes a central role, primarily owing to its capacity to bind ferric ions.

In a randomized controlled trial, no safety issues were observed. Liver enzyme levels were monitored, and no anomalies were found. Overall, there were no adverse effects associated with lactoferrin [149].

### 6.3. Butyrate

Butyrate, a four-carbon short-chain fatty acid, derives principally from the fermentation of undigested carbohydrates, particularly resistant starch and dietary fiber, and, to a lesser degree, dietary and endogenous proteins [150], by the gut microbiota. The gut microbiota generates around 500–600 mmol of short-chain fatty acids (SCFAs), with butyrate estimated to contribute to 20% of the overall SCFA production [151]. SCFAs are absorbed in both the small and large intestines through similar mechanisms, which involve the diffusion of the undissociated form and the active transport of the dissociated form facilitated by SCFA transporters [152,153]. The ionized form of butyrate is actively transported across the apical surface of intestinal epithelial cells via H+-monocarboxylate transporter-1 (MCT1) [154]. The absorbed butyrate is metabolized in the intestinal epithelial cells, liver cells, and other tissues and cells [155]. Within the epithelial cells, butyrate undergoes conversion into acetyl-CoA and enters the tricarboxylic acid (TCA) cycle within the mitochondria to generate ATP, serving as an energy source for colon epithelial cells.

In addition to its energy function for colonocytes, butyrate has garnered significant attention due to its wide-ranging benefits, such as playing a crucial role as a mediator in anti-inflammatory and antitumorigenic activities [156]. Butyrate functions by either inhibiting histone deacetylase (HDAC) or signaling through various G protein-coupled receptors (GPCRs). Its recent attention stems from its advantageous impacts on intestinal homeostasis and energy metabolism. With its anti-inflammatory attributes, butyrate improves intestinal barrier function and mucosal immunity [157].

Butyrate interacts with G protein-coupled receptors such as GPR41, GPR43, and GPR109A, which are found on the surface of intestinal epithelial cells, adipocytes, and immune cells. The activation of GPR41 by butyrate regulates body energy expenditure and upholds metabolic balance [158]; meanwhile, GPR109A (HCAR2) triggers signaling that activates the inflammasome pathway in colonic macrophages and dendritic cells. This activation leads to the differentiation of regulatory T-cells and T-cells producing IL-10, functioning as an anti-inflammatory and anticancer agent in the colon [159].

Butyrate can function as a modulator of the chemotaxis and adhesion of immune cells [160]. It has the ability to influence the migration of neutrophils to inflammatory sites mediated by intestinal epithelial cells and engages various signaling pathways in both gut immune cells and epithelial cells to restore impaired colonic barrier function and maintain gut homeostasis. The heightened reassembly of tight junctions (TJs) and the restoration of transepithelial electrical resistance (TER) were attributed to the activation of AMP-activated protein kinase (AMPK) induced by butyrate [161]. Additionally, butyrate stimulates the production or release of chemokines in neutrophils, dendritic cells (DCs), and endothelial cells, thereby regulating the recruitment of leukocytes [162]. In addition to its effects on cells, these microbial metabolites inhibit HDAC (histone deacetylase) activities both in vitro and in vivo [163,164]. However, the mechanism behind this remains unclear. Butyrate’s effects, primarily mediated through HDAC inhibition, encompass the inhibition of cell proliferation, induction of cell differentiation or apoptosis, and modulation of gene expression [165], acting as an antitumor agent. In addition to this role, butyrate also contributes to anti-inflammatory effects, partially through HDAC inhibition. Multiple human and animal studies have indicated that butyrate downregulates proinflammatory cytokines such as IFN-γ, TNF-α, IL-1β, IL-6, and IL-8 while simultaneously upregulating IL-10 and TGF-β.

Postbiotics show considerable promise as supplements for human health. Specifically, their beneficial impact on microbiota development, intestinal maturation, and various immunomodulatory effects make them particularly intriguing for children, as childhood represents a critical window of opportunity for long-term health. Butyrate shows promise as a valuable intervention for improving the health of newborns and premature infants, potentially preventing serious conditions like necrotizing enterocolitis [166] and late-onset sepsis by enhancing the integrity of the intestinal barrier through promoting the production of the characteristic mucins of the intestinal epithelium (MUC2). Moreover, obesity is increasingly prevalent among children globally, and butyrate could offer a promising supplement in combating this widespread epidemic. In a randomized clinical trial by Coppola et al., researchers discovered that supplementation with butyrate reduced HOMA-IR and fasting insulin levels in children with obesity. Additionally, the analysis of the gut microbiota supported butyrate’s role in glucose metabolism, with children showing a more favorable response if they had a higher abundance of butyrate-producing bacteria at the beginning of the study [167].

In summary, appropriate concentrations of butyrate aid in preserving intestinal barrier function and modulating the immune response within the gut. Both clinical trials and animal studies have demonstrated that butyrate has the potential to alleviate mucosal inflammation and enhance barrier function. Hence, formulations of butyrate and butyrogenic compounds could offer alternative therapeutic strategies for various diseases.

In pediatric patients, butyrate administration has been associated with some adverse effects. Specifically, transient mild nausea and headaches have been observed as side effects [167].

## 7. Conclusions

Products known as oral nutritional supplements (ONSs) are FSMPs (Food for Special Medical Purposes) intended for the prevention or management of caloric-protein malnutrition. They are available in liquid, creamy, or powder form for individuals who can still feed themselves naturally. ONSs are currently available in a wide range of nutritional variants with either standard or disease-specific formulations, as per the document prepared by the European Society of Clinical Nutrition and Metabolism (ESPEN).

Overall, the systematic review revealed evidence supporting the significant role of ONSs in pediatric nutrition. Many studies reported significant improvements in patient outcomes, particularly in clinical nutrition, which provides therapeutic benefits.

The evidence suggests that integrating ONSs with routine medical care can enhance treatment outcomes for children with specific clinical conditions whose dietary management cannot be achieved through natural food modifications alone.

However, the heterogeneity of the studies limits the generalizability of the results. The importance of thorough nutritional assessments, early detection of malnutrition, and addressing nutritional deficiencies is underscored to prevent complications such as muscle weakness, immune deficits, increased infection susceptibility, microbiota–immune system dysregulation, and severe growth and developmental delays. According to the European Society for Paediatric Gastroenterology, Hepatology, and Nutrition (ESPGHAN), “insufficient oral intake” is defined as failing to meet 60–80% of nutritional needs for more than 10 consecutive days in children over one year of age, necessitating timely supplementation.

For children aged one year and older, ONSs are recommended to counteract protein-calorie malnutrition when oral feeding is inadequate. In children with fragile or impaired digestive systems, easily digestible supplements are preferred to prevent and manage malnutrition. This review broadens the definition of “malnutrition” to include selective deficits of bioactive molecules.

This review also explores the growing interest in incorporating bioactive compounds into ONSs, tailored to meet the specific nutritional needs of children. These bioactive compounds exhibit a range of benefits, including antioxidant, anti-inflammatory, and immunomodulatory properties. Understanding the mechanisms of these compounds is crucial for leveraging their potential in pediatric nutrition.

Key conclusions from this systematic narrative review include:

Enhanced nutritional profiles: bioactives contribute to the overall nutritional value of oral supplements, ensuring that children receive a balanced intake of essential nutrients. This is particularly crucial for children with specific dietary restrictions or health conditions that impair nutrient absorption. Healthcare providers should consider personalized nutrition plans that incorporate specific bioactives based on individual patient needs. For example, children with a history of gastrointestinal issues may benefit more from probiotics and prebiotics, while those with bone health concerns may require increased vitamin D supplementation.

Immune support: Several bioactives have been shown to bolster the immune system, reducing the incidence and severity of infections in children.

Growth and development: Omega-3 fatty acids and other essential nutrients play a critical role in the cognitive and physical development of children. Bioactive-enriched supplements can support optimal growth trajectories and cognitive function.

Gut health: Probiotics and prebiotics are instrumental in maintaining a healthy gut microbiota, which is linked to improved digestion, enhanced nutrient absorption, and overall gastrointestinal health.

Disease prevention and management: Bioactives have therapeutic potential in managing chronic pediatric conditions such as obesity, diabetes, and gastrointestinal disorders. They offer a non-pharmacological approach to disease management, which is particularly advantageous in pediatric care.

Safety and efficacy: This review highlights the importance of rigorous clinical trials to establish the safety and efficacy of bioactives in pediatric populations. While current evidence is promising, more research is needed to confirm their long-term benefits and potential risks. In summary, oral nutritional supplements containing bioactives are generally well-tolerated. While minor and transient side effects may occur, serious adverse effects are rare. Nevertheless, monitoring and proper dosing are essential to minimize potential risks.

Personalized nutrition: The inclusion of bioactives in supplements allows for a more tailored approach to pediatric nutrition, addressing individual health needs and dietary gaps. This personalized strategy can lead to better health outcomes and enhanced quality of life for children.

Regulatory and quality control: Ensuring the quality, purity, and consistency of bioactive-enriched supplements is crucial. Regulatory frameworks need to adapt to the growing use of bioactives to guarantee safety and efficacy standards are met.

However, further research and clinical validation are needed to fully establish the efficacy and safety of biocompound-enriched ONSs in pediatric populations. Additionally, practical considerations such as taste, texture, and palatability must be addressed to ensure acceptance and compliance among young patients.

Future advancements in this field will require collaboration among healthcare professionals, researchers, and industry partners. Through combined efforts and resources, continued innovation and refinement of ONS formulations can more effectively meet the nutritional needs of children globally.

## Figures and Tables

**Figure 1 nutrients-16-02067-f001:**
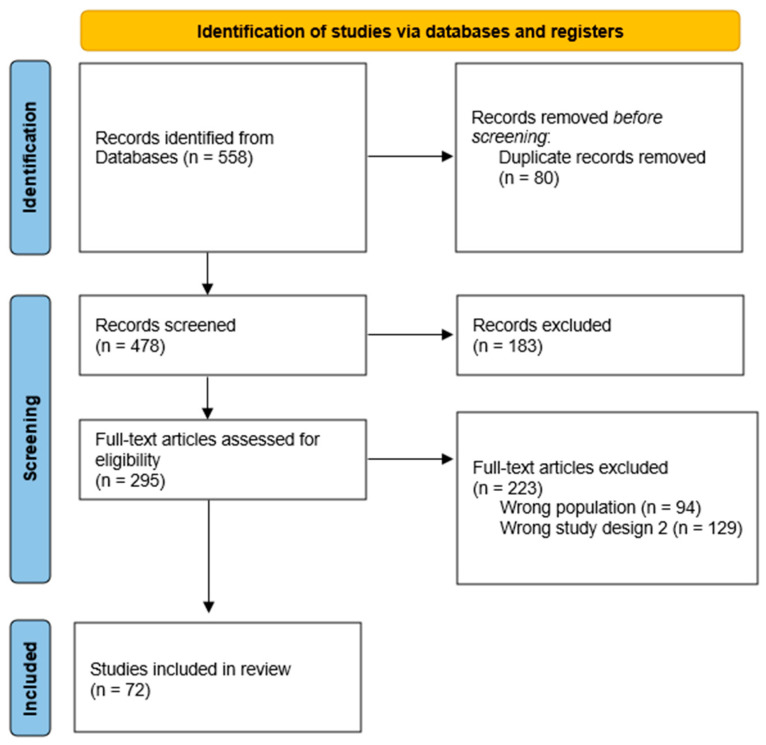
PRISMA flow chart. An overview of the employed systematic search strategy.

**Figure 2 nutrients-16-02067-f002:**
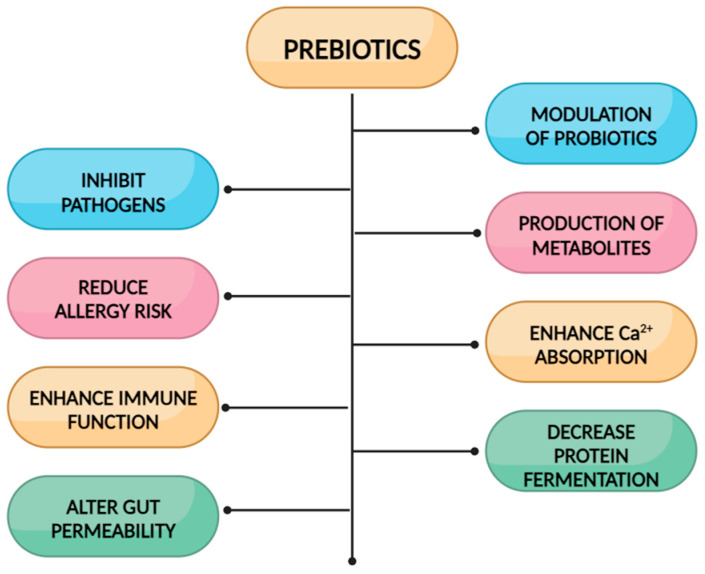
The effects of prebiotics on pediatric health outcomes. Prebiotics contribute to various health benefits by inhibiting pathogens, reducing allergy risk, enhancing immune function, and altering gut permeability. They also modulate probiotics, produce beneficial metabolites, enhance calcium absorption, and decrease protein fermentation. These multifaceted effects highlight the importance of prebiotics in oral nutritional supplements for pediatric populations.

**Figure 3 nutrients-16-02067-f003:**
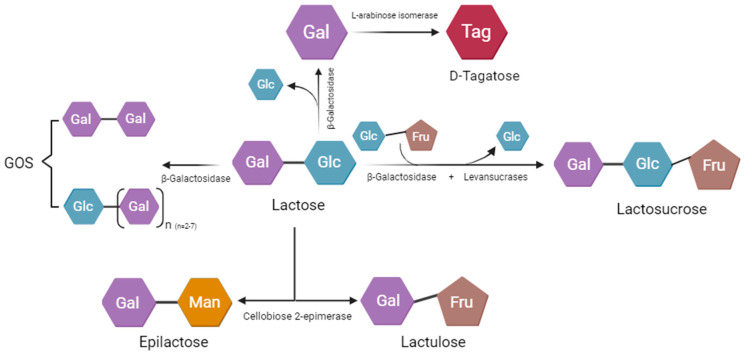
An overview of the enzymatic production of functional lactose derivatives.

**Figure 4 nutrients-16-02067-f004:**
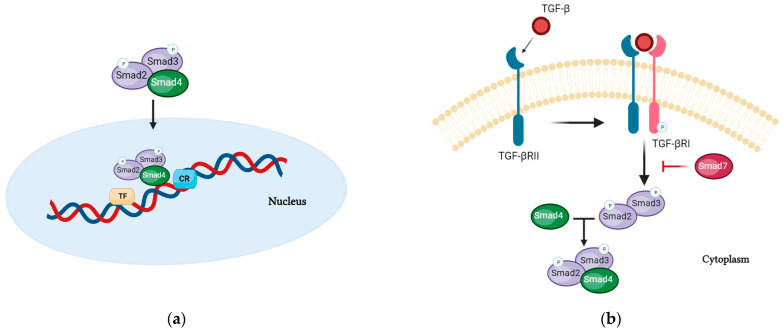
(**a**). TGF-β intracellular signaling through the Smad-dependent canonical pathway. TGF-β binds to transmembrane type I and type II receptors. Receptor-regulated SMADs associate with the common mediator SMAD4 protein and form trimeric complexes. (**b**). The SMAD complex is shuttled to the nucleus, where it binds to specific DNA sequences and cooperates with DNA-binding transcription factors and chromatin modifiers to regulate the expression of TGF-β-responsive genes.

**Figure 5 nutrients-16-02067-f005:**
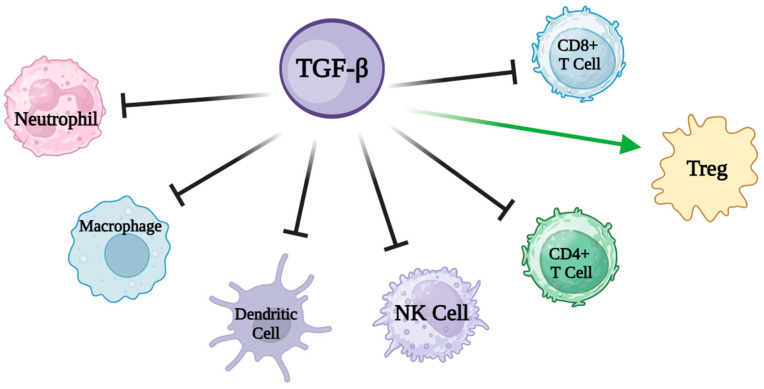
TGF-β’s effects on different cells of the immune system. TGF-β suppresses all kinds of cells of the immune system, such as CD8(+) and CD4(+) T-cells, NK cells, dendritic cells, macrophages, and neutrophils. In addition, TGF-β induces Treg cells.

**Figure 6 nutrients-16-02067-f006:**
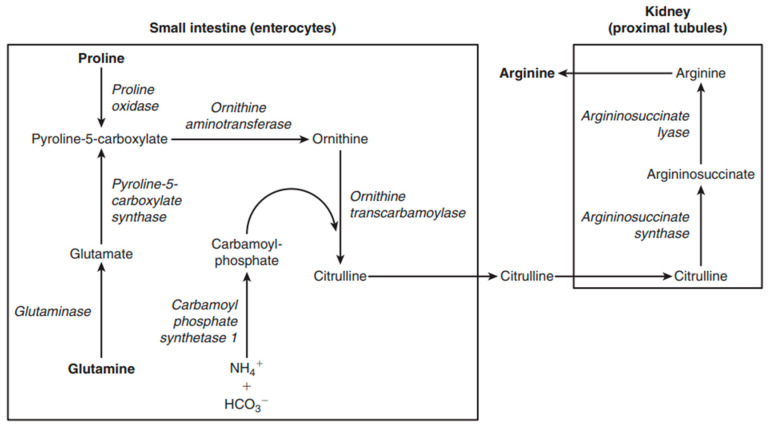
Intestinal–renal axis for the endogenous synthesis of arginine in adult animals.

**Figure 7 nutrients-16-02067-f007:**
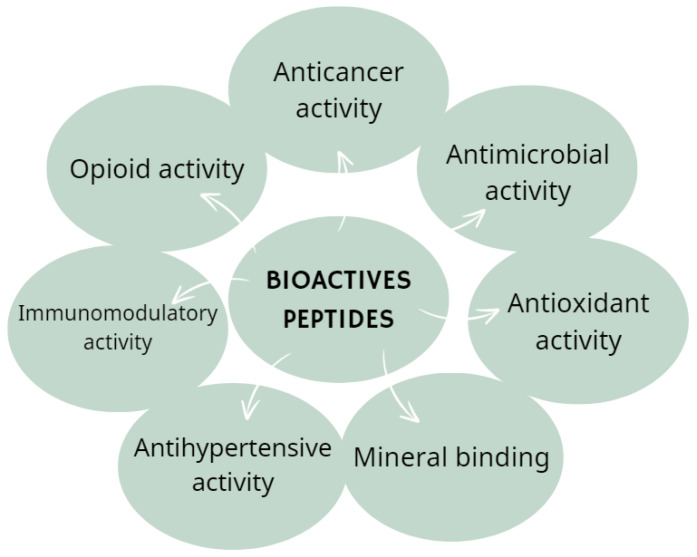
Therapeutic potential of bioactives peptides.

**Figure 8 nutrients-16-02067-f008:**
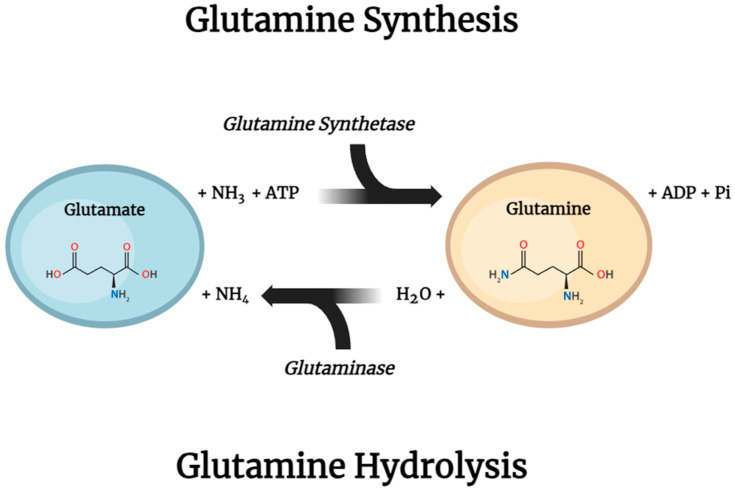
Glutamine metabolism.

**Figure 9 nutrients-16-02067-f009:**
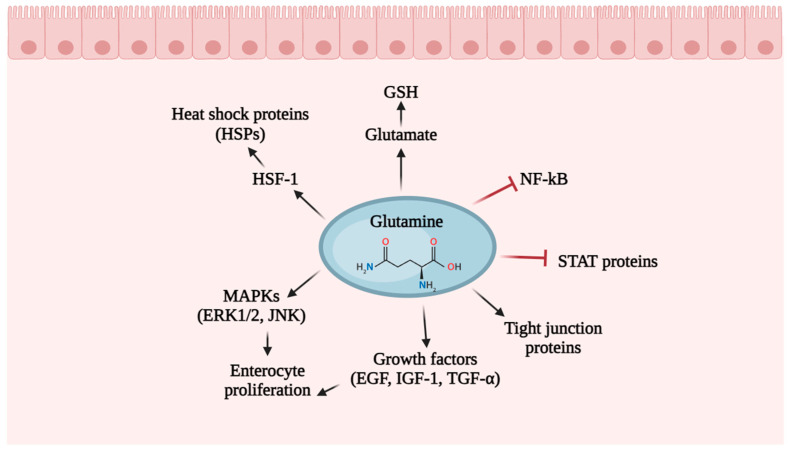
Glutamine’s functions.

**Table 1 nutrients-16-02067-t001:** Summary of characteristics of oral nutritional supplements and recommendations for use.

Formula Type	Summary of Characteristics	Recommendations for Use
Polymeric	Comprise macronutrients such as unhydrolyzed proteins, fats, and carbohydrates. Varies in concentration within the range of 1–2 kcal/mL. Include essential vitamins and minerals. Might be tailored for specific diseases and/or incorporate prebiotics and probiotics.	Designed for patients without severe malabsorptive disorders.
Fiber-containing	The inclusion of fiber is meant to enhance the well-being of the gastrointestinal tract by regulating the frequency and/or consistency of stool through the maintenance of a healthy GI flora. The fiber content generally falls below the recommended total daily fiber intake. These products may include prebiotics in the form of fructooligosaccharides, oligofructose, or inulin. May incorporate probiotics.	Suggested for patients experiencing diarrhea and/or to support or sustain the gut microbiota.
Whole food/blenderized	Whole foods blended to create a texture suitable for enteral consumption, enabling patients to obtain nutritional qualities not present in typical enteral formulas, such as phytochemicals.	Only deemed suitable for application in medically stable patients with a healed feeding tube site and no indications of infection. Should be administered as bolus feeds to uphold safe food practices. Involvement of a dietitian is crucial in formulating the feeding composition to ensure sufficient nutrient delivery.
Diabetes/glucose intolerance	Designed to alleviate hyperglycemia by incorporating a macronutrient composition of 40% carbohydrates, 40% fats, and 20% protein. The presence of fats and soluble fibers in the diet may impede gastric emptying, thereby averting an increase in blood glucose levels.	Current research does not strongly support the use of enteral formulas specifically designed for individuals with diabetes mellitus (DM). Instead, emphasis should be placed on avoiding overfeeding.
Renal	Fluid restricted. Designed with reduced levels of electrolytes, particularly potassium and phosphorus, to avoid excessive supply in patients with renal insufficiency. Protein content may vary.	The initial choice for patients with renal insufficiency should be a standard enteral formula. If there are notable electrolyte imbalances or they arise, contemplation of a renal formula is warranted until electrolyte levels stabilize.
Hepatic	Formulated with reduced protein content featuring a higher proportion of branched-chain amino acids and lower aromatic amino acids to mitigate the risk of hepatic encephalopathy. The lower protein content, however, may lead to insufficient protein delivery. Designed with restrictions on fluid and sodium to lessen the impact of ascites.	As the initial approach, the standard enteral nutrition (EN) formula is recommended for patients with hepatic encephalopathy. Reserve only for individuals with encephalopathy where conventional treatment with luminal-acting antibiotics and lactulose fails to improve the condition.
Elemental/semielemental	Macronutrients are hydrolyzed to maximize absorption.	The objective is to limit enteral delivery to 60–70% of the target energy requirements while ensuring sufficient protein intake. Designed for individuals with malabsorptive disorders; not recommended for regular use.
Pulmonary/fish oil	In efforts to reduce carbon dioxide production, these formulas contain >50% total calories from fat, with lower carbohydrate (<30%) and similar protein content (16–18%). Typically, they also contain ω-3 fatty acids derived from fish oil to increase the delivery of the anti-inflammatory properties of EPA/DHA.	Implement measures to avoid excessive enteral nutrition (EN) delivery, aiming to minimize complications linked to overfeeding. Exercise caution when considering the use of pulmonary formulas in septic and critically ill patients.
Immunonutrition/ immunomodulating	Contain pharmacologically active substances, such as arginine, glutamine, ω-3 fatty acids, γlinolenic acid, nucleotides, and/or antioxidants in efforts to modulate immune function.	Providing immune-modulating substances as part of enteral nutrition (EN) may offer potential benefits for patients undergoing elective surgery. However, the existing research is insufficient to endorse the routine use of immune-modulating formulas for critically ill patients.

**Table 2 nutrients-16-02067-t002:** Summary of characteristics of prebiotics and recommendations for use [11,18].

Prebiotic	Health Benefits
FOS	Enhances gut health by fostering the proliferation and function of beneficial gut bacteria; aids in regulating blood sugar levels by delaying carbohydrate digestion; potentially lowers the risk of specific cancers; may contribute to better bone health through enhanced calcium absorption; and could diminish the risk of heart disease by lowering cholesterol levels and inflammation.
GOS	Enhances gut health by stimulating the growth and function of beneficial gut bacteria; alleviates the risk of constipation and related digestive problems by promoting regular bowel movements; boosts the immune system through heightened production of beneficial bacteria; lowers the likelihood of infections and inflammatory bowel conditions; and may aid in weight management by enhancing satiety and reducing calorie consumption.

**Table 3 nutrients-16-02067-t003:** The role of TGF-β present in ONSs.

**ONSs Supplemented with TGF-β**	**Disease**	**Functions**
	IBD	Immunomodulatory roles especially relevant to the GI tract.
		Healing of the intestinal mucosa and in the development of fibrosis and stenosis through the regulation of fibroblast activity.

**Table 4 nutrients-16-02067-t004:** Bioactive nature of the components and their roles within oral nutrition supplements targeted at enhancing physiological functions in pediatric populations.

Oral Nutrition Supplements	Characteristics	Where Is It?	Functions
**Arginine**	Conditionally essential amino acid in situations such as developmental stages and certain pathological conditions (infection or inflammation, or under conditions of impaired renal and/or intestinal metabolic functions), where endogenous production is insufficient.	Seafood, watermelon juice, nuts, seeds, algae, meats, rice protein concentrate, and soy protein isolate.	- Plays a role in immunoregulation. - Precursor of proteins, nitric oxide, proline, creatine, agmatine, and polyamine. - Induces the secretion of hormones such as insulin, glucagon, growth hormone, and prolactin.
**Omega-3**	Are polyunsaturated fatty acids (PUFAs) characterized by a double bond at the *n*-3 position. These PUFAs include eicosapentaenoic acid (EPA), alpha-linolenic acid (ALA), and docosahexaenoic acid (DHA). Alpha-linolenic acid is essential (ALA).	EPA and DHA are mainly found in the marine ecosystem, fish and seafood, but also in leafy vegetables and nuts, which are famous for their high *n*-3 FA content, making them the main source of ALA. Leafy vegetables and nuts are also famous for their high *n*-3 FA content, making them the main source of ALA.	- Affects the structure and function of biological membranes, such as elasticity, membrane organization, and ion permeability. - Reduces the production of inflammatory cytokines TNFα and IL-6.
**Nucleotides**	Consist of a sugar molecule (ribose in RNA or deoxyribose in DNA), a phosphate group, and a nitrogen-containing base. The bases found in DNA are adenine (A), cytosine (C), guanine (G), and thymine (T), while RNA substitutes uracil (U) for thymine. Both DNA and RNA molecules are polymers composed of extended chains of nucleotides.	Nucleotides are obtained in the diet and are also synthesized from common nutrients by the liver. Naturally found in all foods of both animal and plant origin (healthy individuals typically consume 1–2 g of nucleotides daily through their diet), or through endogenous synthesis, which serves as the primary source of nucleotides.	- Play a fundamental role in organismal physiology, serving as the fundamental units of nucleic acids and repositories of chemical energy. - Carriers of activated metabolites for biosynthesis, structural components of coenzymes, and regulators of metabolism.

## Data Availability

Not applicable.

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
