# Peer review of "Bioactives in Oral Nutritional Supplementation: A Pediatric Point of View"

_nutrients, 2024, doi:10.3390/nu16132067_

Round 1

Reviewer 1 Report (Previous Reviewer 1)

Comments and Suggestions for Authors

1. Clarity of research background and purpose: This study outlines the significance of oral nutritional supplements in pediatric populations and proposes the objective of assessing the effects of bioactive compounds. It is recommended that the inadequacy of existing nutritional supplements to meet this need is further clarified, as well as the selection criteria and expected actions of the newly added bioactive compounds.
2. Detailed description of the quality assessment of the included literature and the data extraction process is also needed.
3. Completeness and transparency of results: While the potential benefits of various bioactive compounds have been mentioned, it is advised to provide detailed mechanisms of action of these compounds and their specific impacts on health outcomes, including dose-response relationships and dose sensitivity analysis.
4. Depth and breadth of discussion: The study discusses the prospects of bioactive compounds in ONS applications but lacks an evaluation of the quality of evidence and consistency of research in the existing literature. It is suggested that discussion on the current limitations of the research and future directions should be added, such as the long-term safety and efficacy of bioactive components.
5. Practicality of conclusions and recommendations: The paper's conclusions underscore the potential value of ONS in pediatric nutrition, but it is suggested that further specific clinical application recommendations are proposed, including how to integrate into existing nutritional supplement schemes and how to design future clinical trials to validate these conclusions.6. Innovation of the study: Given the existing foundation of ONS research, authors should highlight the innovation of their work in the paper, demonstrating the added value of this review to the existing knowledge base.

Author Response

1: Thank you for your assessment. We believe that conducting additional clinical studies beyond those already selected and specified in the study is essential to achieve a thorough and dependable understanding of the effects of bioactive compounds in oral nutritional supplements. These studies should focus on evaluating the long-term safety, efficacy, and specific mechanisms of action of bioactive compounds, as well as dose-response relationships. Only through well-designed and rigorous clinical studies can we obtain solid evidence to optimize nutritional supplement formulations and ensure they effectively meet the nutritional needs of children.

2: To ensure the transparency and reliability of the study results, we can implement the review to assess the quality of the selected studies.

3: If necessary, the section on the mechanisms of action of the mentioned biochemical compounds will be included in the work. However, there are not enough clinical studies to explain in detail their impacts on the health of the pediatric population.

4: We will add a detailed discussion on the current limitations of the research and future directions, including the long-term safety and efficacy of bioactive components. This exploration will help clearly outline current challenges and identify priority areas for future research.

5: Certainly, as mentioned earlier, it is essential to conduct additional clinical studies to gain a better understanding of the topic. These studies will provide a stronger foundation for proposing specific recommendations for clinical application, including methods to integrate these compounds into existing nutritional supplement regimens and to design future clinical studies that can validate these conclusions.

6: Certainly, we can highlight more explicitly how our study may bring new perspectives or approaches that enrich the research field on ONS supplemented with bioactive compounds in pediatrics.

Reviewer 2 Report (Previous Reviewer 2)

Comments and Suggestions for Authors

Since I reviewed the previous version, I shall not repeat my previous comments here. Regarding this new version of a review of bioactives in oral nutritional paediatric supplementation, it is pleasing to see that the authors have followed my advice in full. The PRISMA guidelines have been followed. There is now a flow chart illustrating a systematic search strategy. The data sources and study selection criteria are detailed.

In sum, this is a much improved version.

Author Response

Thank you for your thoughtful and positive feedback on our revised manuscript. We greatly appreciate your previous comments and are pleased to hear that you find the updated version much improved.

 Additionally, we have detailed the data sources and study selection criteria to enhance the transparency and rigor of our review.

Your guidance has been invaluable in improving the quality of our work, and we are grateful for your continued support. Thank you once again for your insightful review.

Reviewer 3 Report (New Reviewer)

Comments and Suggestions for Authors

Dear authors,

congratulations on the interesting topic.

I have several important remarks:

1. The manuscript is not prepared according to MDPI guidelines (check references, fonts, etc)

2. Lines 46-54. "Oral Nutritional Supplements (ONS) are part of the Enteral Nutrition (EN) products 46 designed to deliver both macro and/or micro nutrients. EN encompasses various forms 47 of nutritional support involving the utilization of "dietary foods for special medical 48 purposes" as defined by the European legal regulation in Commission Directive 49 1999/21/EC of 25 March 19991, irrespective of the administration route."

The term "Oral Nutritional Supplements" is adopted mainly in UK.

https://www.bapen.org.uk/education/nutrition-support/nutrition-by-mouth/oral-nutritional-supplements-ons/#:~:text=Oral%20Nutritional%20Supplements%20(ONS)%20are,requirements%20through%20oral%20diet%20alone.

In the European Union is adopted the term "Food supplements" (or dietary supplements).

https://www.efsa.europa.eu/en/topics/topic/food-supplements

https://food.ec.europa.eu/safety/labelling-and-nutrition/food-supplements_en

In my view in this paragraph you should mention that Oral Nutritional Supplements and Food supplements refer to the same term. You should include data about the legislation of these products in Europe, USA, or other countries.

The legislation (regulation, introduction, quality control) about these products is quite different in the UK and in the  European Union (you should include some reference and to describe this).

The differences in legislation highly affect the the quality of these products.

You did not mention anything about the legislation and the quality control of these products.

3. Include references in table 2.

4. Page 15. You should give a name to this table (not just box 2). Include references in the table, as well.

5.You should include a section which discuss the "Health Risk and side effects of the supplementation".  In general, in the last decade many studies reported  many undeclared compounds.

You can check this issue in Goggle scholar or Pubmed. Just write the key words "dietary supplements undeclared compounds" or "food supplements undeclared compounds".You will see many studies which discuss this issue.  The poor quality of  many food supplements  is a results of the mild legislation. However, the presence of undeclared compounds (normally pharmacologically active compounds) can seriously affect children health. That is why you should discuss this current problem.

6. A conclusion section must be included. The conclusions must be reformulated and explained more clear.

Author Response

Dear Reviewer, thank you for your detailed comments and for taking the time to review our manuscript. 

  1. We would like to clarify that we have utilized the Nutrients template as recommended by the MDPI guidelines. . We will have a double-check to ensure they are in the correct format and style.
  2. Regarding lines 46-54, we will revise the text to clarify the terminology. We refer to the ESPEN Guidelines 2017, specifically sections 3.6.1 and 3.6.2, which cover Nutritionally Complete and Nutritionally Incomplete Oral Nutritional Supplements (ONS). While both ONS and food supplements aim to improve nutritional intake, ONS are typically intended for individuals with specific medical needs and are often used under medical supervision, whereas food supplements are more broadly aimed at enhancing the general population's diet. As you suggested, we will certainly improve the introduction to reflect these distinctions.
  3. We will include the necessary references in Table 2. We will ensure all information is properly cited.
  4. We will give a proper title to the table currently labeled as "Box 2". As reference the Box has been created by Professor Maria Immacolata Spagnuolo
  5. We agree that addressing this issue is crucial, especially considering its implications for children's health. Although our search on Google Scholar using terms "ONS undeclared compounds" and similar but not find anything concern the specific case
  6. We will revise the conclusion section to ensure it is comprehensive, clear, and summarizes the key findings effectively.

We will make the necessary revisions to address these points and will resubmit the manuscript for your review.

Thank you once again for your insightful comments.

Round 2

Reviewer 3 Report (New Reviewer)

Comments and Suggestions for Authors

The authors did not take in mind my remarks.

Author Response

from round 1 

COMMENT 5: You should include a section which discuss the "Health Risk and side effects of the supplementation".  In general, in the last decade many studies reported  many undeclared compounds. You can check this issue in Goggle scholar or Pubmed. Just write the key words "dietary supplements undeclared compounds" or "food supplements undeclared compounds".You will see many studies which discuss this issue.  The poor quality of  many food supplements  is a results of the mild legislation. However, the presence of undeclared compounds (normally pharmacologically active compounds) can seriously affect children health. That is why you should discuss this current problem

RESPONSE  5.:  Thank you for your valuable feedback on our manuscript. We have implemented the requested changes and added information on the side effects of bioactives included in oral nutritional supplements. Below are the specific updates made:

  1. Fiber: at line 186.
  2. TGF-beta:  at line 395.
  3. Arginine: I at line 539.
  4. N-3 PUFA: at line 590.
  5. Lactoferrin: at line 866.
  6. Butyrate:  at line 935.
  7. Conclusion: Summarized the side effects and emphasized the importance of monitoring and managing these when including bioactives in oral nutritional supplements at line 989.

To address concerns about the presence of undeclared compounds in oral nutritional supplements (ONS), it is important to note that rigorous quality control measures are in place for these products. Comprehensive testing and regulatory standards ensure that ONS do not contain undeclared compounds.

We hope these revisions meet your expectations and provide a comprehensive overview of the side effects associated with the bioactives. Thank you for your continued guidance and support.

This manuscript is a resubmission of an earlier submission. The following is a list of the peer review reports and author responses from that submission.

Round 1

Reviewer 1 Report

Comments and Suggestions for Authors The article "Bioactives in Oral Nutritional Supplementation: a pediatric point of view" is an interesting study. It is particularly important for the development of cognitive function in children and adolescents. The overall design of the study is reasonable and logical, and it has important practical significance and value in the healthy development of current children and adolescents. However, there are some problems with the manuscript that need to be revised and improved.

Firstly, the abstract of the manuscript does not seem to meet the publication requirements of the journal and does not describe the specific findings and conclusions of the study. Further refinement of the abstract section of the manuscript is needed.

Secondly, the research methodology section of this manuscript, as a review paper, does not provide enough detail about the extracted articles. This is particularly important for the final results of this study.

Thirdly, it affects to list in detail the detailed information of the review articles included in this study, as well as the related detailed criteria for inclusion and exclusion.

Fourthly, the discussion section of the manuscript does not go into enough depth and the appropriate references should be added to analyse again in depth the specific reasons that led to the results of this study.

Fifth, the reference format of the manuscript is not agreeable enough and needs further improvement.

Sixth, the images in the manuscript appear to be AI-generated images. It raises some doubts about the originality.

Reviewer 2 Report

Comments and Suggestions for Authors

The authors have reviewed the area of paediatric oral nutritional supplementation. Clinically, this is an important area to review. They have covered a wide range of supplements in a far reaching article.

Overall, this is simply a verbose narrative review. Importantly, there is no methods section. Thus, if another researcher were to be asked to produce a review of the same area, it is highly probable that the new researcher would not replicate the key points from the present submission. Even some of the conclusions of individual sections would differ.

I strongly recommend that the authors completely re-write their submission, this time making it a systematic review. As part of this process, for each major supplement, the authors should list the databases and registers from which relevant studies were identified. The screening process for each such set of studies, including the eligibility criteria, should be detailed. A flow diagram for each supplement should be included in the new submission. Each such diagram should give the numbers of databases and registers identified; the numbers removed before screening; the number of studies screened; the number excluded; the number sought for retrieval and the number not retrieved; the number assessed for eligibility; the number excluded, including the reasons for excluding each of these; and the numbers included in the review of that particular supplement.